# Silver Nanoparticle-Anchored Human Hair Kerateine/PEO/PVA Nanofibers for Antibacterial Application and Cell Proliferation

**DOI:** 10.3390/molecules26092783

**Published:** 2021-05-08

**Authors:** Jiapeng Tang, Xiwen Liu, Yan Ge, Fangfang Wang

**Affiliations:** 1Department of Physiology and Hypoxic Biomedicine, Institute of Special Environmental Medicine, Nantong University, Nantong 226019, China; jptang@ntu.edu.cn (J.T.); 1925310015@stmail.ntu.edu.cn (X.L.); 2Co-Innovation Center of Neuroregeneration, Nantong University, Nantong 226001, China; 3School of Textile and Clothing, Nantong University, Nantong 226019, China; 4National & Local Joint Engineering Research Center of Technical Fiber Composites for Safety and Protection, Nantong University, Nantong 226019, China; 5College of Fine Arts and Design, Yangzhou University, Yangzhou 225009, China; wffsjj@ytu.edu.cn

**Keywords:** silver nanoparticles, kerateine anchoring, nanofibers, antibacterial activity, cell proliferation

## Abstract

The main core of wound treatment is cell growth and anti-infection. To accelerate the proliferation of fibroblasts in the wound and prevent wound infections, various strategies have been tried. It remains a challenge to obtain good cell proliferation and antibacterial effects. Here, human hair kerateine (HHK)/poly(ethylene oxide) (PEO)/poly(vinyl alcohol) (PVA) nanofibers were prepared using cysteine-rich HHK, and then, silver nanoparticles (AgNPs) were in situ anchored in the sulfur-containing amino acid residues of HHK. After the ultrasonic degradation test, HHK/PEO/PVA nanofibrous mats treated with 0.005-M silver nitrate were selected due to their relatively complete structures. It was observed by TEM-EDS that the sulfur-containing amino acids in HHK were the main anchor points of AgNPs. The results of FTIR, XRD and the thermal analysis suggested that the hydrogen bonds between PEO and PVA were broken by HHK and, further, by AgNPs. AgNPs could act as a catalyst to promote the thermal degradation reaction of PVA, PEO and HHK, which was beneficial for silver recycling and medical waste treatment. The antibacterial properties of AgNP-HHK/PEO/PVA nanofibers were examined by the disk diffusion method, and it was observed that they had potential antibacterial capability against Gram-positive bacteria, Gram-negative bacteria and fungi. In addition, HHK in the nanofibrous mats significantly improved the cell proliferation of NIH3T3 cells. These results illustrated that the AgNP-HHK/PEO/PVA nanofibrous mats exhibited excellent antibacterial activity and the ability to promote the proliferation of fibroblasts, reaching our target applications.

## 1. Introduction

Keratins, a family of fibrous proteins, are abundant in fibers and hard biostructures. A high level of cysteine residues (7–20 mol%) in the peptide chain of keratin form many intermolecular and intramolecular disulfide bonds in their natural form. Therefore, keratin has stable chemical properties [1,2]. The spatial structure of keratin has two types: an α-helical structure and β-sheet structure [3]. The former, containing a 36% cystine content, exist in the stratum corneum, wool, hair, quills, horns, hooves, nails, whale baleen, hagfish slime thread and whelk egg capsules, and the latter, having up to 2% cystine residue, are mainly included in feathers, beaks and claws [4]. Additionally, keratins contain cell adhesion motifs similar in structure to extracellular matrix proteins (such as collagen or fibronectin), arginine-glycine-aspartate (RGD) and leucine-aspartate-valine (LDV), which can support cell attachment and proliferation [5]. These unique structures and biological properties make keratin the focus of the biomedical field, including wound dressing, tissue engineering and drug delivery [6,7,8,9,10,11,12]. Nevertheless, the shortcomings of brittleness, poor mechanical properties and processing properties limit the practical use of keratin [13,14]. Synthetic or natural polymers are usually added as plasticizers and crosslinkers to form composite materials to improve these defects of keratin, such as poly(vinyl alcohol) (PVA), polylactic acid, chitosan, gelatin, poly(ethylene oxide) (PEO) and polyurethane [8,11,15,16,17,18,19]. Electrospun nanofibers prepared by blending keratin with PEO and PVA can further promote tissue regeneration, because they simulate the structure of a natural extracellular matrix (ECM) [17,19]. Keratin nanofibers can be hydrolyzed to produce a large amount of nutrient-rich amino acids and are easily contaminated by microorganisms [20]. Thus, they need to be given high antibacterial activity before being used for biomedical purposes.

Various attempts have been made to extract keratin from raw materials of biologics with the cleavage of disulfide bonds. The usual methods are classified into three categories, namely reduction, oxidation and keratinase, to produce kerateine with free thiol groups and keratose with free sulfonic groups, respectively [21,22]. The long production cycle of enzymatic hydrolysis has, so far, limited the development of industrial processes, and keratin extracted with peroxide loses the active thiol groups [21,23]. The thiol groups of kerateine can capture oxygen-free radicals, which is conducive to cell growth [24]. In addition, keratin-based biomaterials as a biosorbent have been widely used to remove heavy metal ions and toxic pollutants in water [25,26]. In particularly, active thiol groups in kerateine play a key role in the process of adsorbing heavy metal ions [27].

Silver nanoparticles (AgNPs) can better attach to microorganisms due to their large surface areas and nano effects, showing a high antibacterial performance [28]. AgNPs adhere to the cytoplasmic membrane and cell wall of microorganisms, causing disruption, penetrating the cell, interacting with sulfur-containing proteins and phosphorous compounds such as DNA in bacterial cells and inducing the generation of reactive oxygen species and free radicals to attack the respiratory chain, leading to cell death [29]. In turn, AgNPs are released from bacterial cells, so their concentrations and properties do not change, thus enhancing their bactericidal persistence. This advantage is not available in biological antibacterial agents, chemical antibacterial agents and some inorganic antibacterial agents [28]. At present, as an antibacterial agent, nano-silver has been used to produce various antibacterial materials. In the case of keratin nanofibers prepared by the AgNP embedding method, most of the AgNPs are wrapped inside the fibers [17,19]. After the burst release of AgNPs on the fiber surface occurs, the AgNPs inside the fibers can only migrate outward slowly, necessarily alleviating the efficient antibacterial effect. If the active groups on the fiber surface can be used to anchor the AgNPs, the bioavailability of the AgNPs will be greatly improved.

Obviously, human hair kerateine (HHK) has better biocompatibility with the human body and the highest concentration of cysteine compared with animal-derived keratin [30]. Therefore, in this study, the reduced kerateine extracted by human hair was blended with PEO/PVA to fabricate AgNP-anchored HHK/PEO/PVA nanofibers by electrospinning and the in situ anchoring method (Figure 1). The surface morphology, nano-silver positioning and thermal properties and antibacterial activity of the nanofibrous mats were characterized to assess the potential of AgNPs in combination with HHK/PEO/PVA nanofibers, which will provide a robust and compatible material for biomedical applications.

## 2. Results and Discussion

### 2.1. Degradation Resistance of AgNP-HHK/PEO/PVA Nanofibers

Nanofibers need to be resistant to chemical and physical degradation during usage, so the degradation resistance of AgNP-HHK/PEO/PVA nanofibers treated with different AgNO_3_ concentrations was firstly investigated. An ultrasound can instantaneously cause temperatures of roughly 5000 °C and pressure of about 500 atmospheres in the local spots of bubbles through the process of acoustic cavitation [31]. Shockwaves and microjets derived from acoustic cavitation can produce high-velocity interparticle collisions to melt most metals in liquid–solid slurries [32]. Therefore, the subtle differences between fibers can be amplified by an ultrasonic test for the degradation behaviors of fibers. Figure 2 and Appendix A show the degradation process of AgNP-HHK/PEO/PVA nanofibers by an ultrasonic treatment in ethanol and the morphology of the nanofibers in the suspension, respectively. The degradation rate of HHK/PEO/PVA nanofibers treated with a 0.005-M silver nitrate solution was the slowest, and the dissociated fibers still maintained a relatively complete form. A part of the AgNPs distributed on the surfaces of fibers and the other AgNPs were diffused, along with the degradation of the fiber surface. HHK/PEO/PVA nanofibers treated with a 0.01-M silver nitrate solution degraded rapidly, and the boundary of the dissociated fibers fused with the external environment. Some AgNPs were scattered on the outside of the fibers, and parts of them were wrapped up in the vesicular structure formed by the polymers. These vesicles might be formed spontaneously by the polymers after the disintegration of the nanofibers. Seriously, the degradation of the HHK/PEO/PVA nanofibers treated with a 0.02-M silver nitrate solution was the most notable. The degradation rate of the fibers reached about 35% after the treatment for 10 min, which was 154% higher than that treated with a 0.005-M silver nitrate solution. The clear profile of dissociated fibers was not seen, indicating that the fibers were not completely degraded. AgNPs were scattered throughout the entire visual field. The degradation resistance of the AgNP-HHK/PEO/PVA nanofibrous mats decreased significantly with the increase of the silver nitrate concentration, which showed that the silver nitrate treatment had a great influence on the structure of the HHK/PEO/PVA nanofibers. Based on the significant differences in the degradation resistance of the three AgNP-HHK/PEO/PVA nanofibers, the HHK/PEO/PVA nanofibers treated with a 0.005-M silver nitrate solution were used in the subsequent experiment.

### 2.2. Element Distribution of AgNP-HHK/PEO/PVA Nanofibers

Figure 3 reveals the elemental distribution on the AgNP-HHK/PEO/PVA nanofibers. The two basic elements of C and O were all uniformly distributed within the nanofibers, located in the fiber skeleton. The element of nitrogen was a characteristic element of HHK, and the methionine and cysteine residues on HHK were labeled with sulfur. The distribution of HHK on the fibers could not overlap with C and O, and the distribution density was not uniform. The reason was that the proteins and peptides owned their secondary structures, such as the α-helix, β-pleated sheet and random coil, which broke the crystalline region formed by intermolecular and intramolecular hydrogen bonds of the other polymers to reduce the crystallinity of the fibers [33]. The distribution of the sulfur elements marked the sulfur-containing amino acid residues in HHK, methionine and cysteine. Seen from the distributions of S and Ag, most AgNPs and S clusters were colocalized (red circle). This illustrated that the interaction between Ag and S was the key factor for the formation and adhesion of AgNPs on the fiber surface. The sulfur atoms of the methionine residues could link to the AgNPs by coordination bonds to improve the adhesive force of the AgNPs on the fiber surface [34]. The thiol group of cysteine could immediately react with the Ag^+^ ions to form silver mercaptide and evolve H^+^ ions [35]. Furthermore, in the presence of excess silver ions, silver mercaptide could form associates that were, successively, from simple (small) to complex (large): fractal cluster, cluster coalescence and a three-dimensional network. Meanwhile, cysteine could also anchor the silver atoms to form silver nanoclusters, whose structures were governed by the anchoring N-Ag, O-Ag and S-Ag bonds, as well as the nonconventional N-H···Ag and O-H···Ag hydrogen bonds in the surroundings [36,37]. The results of the EDS analysis (Table 1) demonstrated that a small amount of sulfur in HHK immobilized a large amount of silver by coordination bonds or the way of thiol–silver nanoclusters to form AgNPs.

### 2.3. Morphological Properties and Diameter Interpretation

The AgNP-HHK/PEO/PVA nanofibers were successfully fabricated using the in situ anchoring method based on electrospun HHK/PEO/PVA nanofibers. Figure 4 shows the microstructure and diameter distribution of the PEO/PVA, HHK/PEO/PVA and AgNP-HHK/PEO/PVA nanofibers. The diameters of the PEO/PVA nanofibers were 150.96 ± 42.34 nm. The diameter distribution was uniform, and their surfaces were smooth, which provided a good basic framework for the subsequent addition of HHK and the silver anchor. After adding HHK, the fiber diameters increased to 182.45 ± 45.96 nm, which were completely in line with the normal distribution. The diameters of the fibers were much smaller than the electrospun keratin/PEO/PVA ternary nanofibers (249.76 ± 38.02 nm) [17]. This might be caused by the differences in the rheology of the three-component polymer solution or the higher electric field strength and smaller jet velocity [38]. Many burrs appeared on the surfaces of the fibers, and these should be HHK tentacles outstretching from the fibers, which can capture and anchor silver. Moreover, the addition of HHK destroyed the original PEO/PVA system to weaken the mechanical strength of the nanofibers so that more fiber fractures appeared in Figure 4b. The diameters of the fibers turned thinner after the silver nitrate treatment—only 120.87 ± 26.21 nm. There were many adhesions in the fibers, making the diameter distribution slightly worse. The burrs on the fiber surface were replaced by considerable beads, which was confirmed as AgNPs in the TEM images (Figure 3). Previous studies have pointed out that the diameters of the electrospun nanofibers could be reduced by 16–36% after adding AgNPs to the keratin/PEO/PVA ternary system [17]. In this study, the fiber diameter was decreased by about 34% after treating with silver nitrate. The higher proportion of HHK in the HHK/PEO/PVA ternary polymer system made the nanofibers more susceptible to the corrosion of acid and alkali than the nanofibers prepared in the literature [17]. These results indicated that the H^+^ ions produced during the silver nitrate treatment and the excess alkali added to neutralize the H^+^ ions had an etching effect on the fiber surface [39].

### 2.4. FTIR Analysis

A FTIR analysis can be employed to confirm intermolecular interactions within nanofibers. Figure 5 represents the FTIR-ATR spectra of the HHK extract, neat PEO/PVA, HHK/PEO/PVA and AgNP-anchored HHK/PEO/PVA nanofibers. The spectra of the HHK extract showed the characteristic absorption bands of peptides, which was clear evidence of the presence of kerateine. The vibrations of the peptide bonds could be attributed to amide A, amide I, amide II and amide III [19]. The broad absorption band at 3279 cm^−1^ was generally associated with amide A and the stretching vibrations of the N-H bond. The amide I bands (the stretching vibrations of C=O) and amide II bands (the bending vibrations of N-H) were observed at 1637 and 1517 cm^−1^, respectively. Additionally, the absorption bands at 1238 and 1389 cm^−1^ belonged to the amide III vibrations, demonstrating the presence of α-helical structure [40]. In the case of the neat PEO/PVA nanofibers, the broad absorption band at 3303 cm^−1^ was assigned as the O-H stretching of PVA. Additionally, the absorption peaks at 842 and 1467 cm^−1^ referred to the O-H and C=O bending vibrations of PVA. A double peak at 2882 and 2860 cm^−1^ represented the stretching vibrations of CH_2_ [41]. The sharp peak at 1096 cm^−1^ was produced by C-O-C stretching [42]. The functional group profile showed bending vibrations at 962 and 1341 cm^−1^, also representing C-H stretching. The incorporation of HHK into the PEO/PVA nanofibers indicated characteristic peaks of amide bonds (the red line of Figure 5), which confirmed that HHK/PVA/PEO nanofibers had been successfully electrospun. The in situ silver nanoparticles deposited on the fiber surface reduced the infrared peak intensity of PEO/PVA and retained the characteristic peaks of HHK. It was illustrated that the silver nanoparticles restricted the vibration of bonds in PEO and PVA, which had an impact on the intermolecular structure. However, all the hydrogen bonds in the α-helix of HHK were intramolecular hydrogen bonds, and they pointed to the same direction along the helix axis, resulting in a stable configuration. The thiol groups of the cysteine residues in HHK interacted with silver ions to form silver thiolate nanoclusters (AgNPs, Figure 3), which would not cause changes in the HHK structure [43].

### 2.5. XRD Analysis

The XRD patterns of the PEO/PVA, HHK/PEO/PVA and AgNP-anchored HHK/PEO/PVA nanofibers were analyzed to assess the crystallinity of the nanofibers and confirm the loading of kerateine on the PEO/PVA nanofibers (Figure 6). The black line corresponds to neat PEO/PVA nanofibers that have two weak peaks (peak1 = 15.1° and peak3 = 23.44°) and a strong intensive peak, peak2, at 19.14°. The three diffraction peaks were considered as the characteristic diffraction peaks of PEO. Peak2 and peak3 of those ones confirmed the presence of the PEO crystalline phases in the blend [17,44,45]. In addition, the diffraction peak at 2θ of 19° was also attributed to the characteristic diffraction peak of PVA [45]. When HHK was added into the spinning solution, the top of peak1 in a blue line became broader, and the intensity of peak2 became weaker, indicating that the PEO crystalline phases were disaggregated into microcrystallines. However, the intensity of peak3 did not change significantly, which suggested that the division of the PEO crystalline phases should be not the main reason for the decrease of the crystallinity. A new interaction between the hydroxyl groups of PVA and side chain groups of various amino acids in HHK emerged to reconstruct the intermolecular hydrogen bonds in the original PVA crystalline phases, which was shown with weak multiple diffraction peaks at 28°, 29° and 33° in Figure 6. Notably, the strong diffraction peak of 19° disappeared almost completely after the nano-silver anchoring, which illustrated that in situ silver anchoring had a great influence on the molecular interactions of PEO/PVA, causing a transformation from the crystalline phase to the amorphous phase in the fibers. Moreover, the intensity and locations of the diffraction peaks at 28°, 29° and 33° also changed, indicating that the formation and anchoring of the AgNPs not only interacted with the active groups on HHK but, also, with the hydroxyl groups on PVA and even the -C-O-C- group in PEO [39].

### 2.6. Thermal Analysis

Figure 7 reveals the DSC and TGA curves of the PEO/PVA, HHK/PEO/PVA and AgNP-HHK/PEO/PVA nanofibrous mats, and the thermal data of DSC and TGA are shown in Table 2. The DSC curves of the PEO/PVA nanofibrous mats exhibited four endothermic peaks (black line). The first endothermic peak corresponded to the melting of the PEO and PVA blended nanofibers manifesting a semi-crystalline structure, which was slightly higher than reported in the literature [46]. It should be due to the larger molecular weight of PEO used in this study [47]. The *T*_m1_ decreased from 47.57 °C to 44.93 °C with the HHK addition. The addition of HHK destroyed the hydrogen bonds of PEO and PVA to decrease the crystallinity of the fibers. The interactions among the macromolecules were weakened, and the Δ*H*_m1_ was also reduced (Table 2). The anchoring of the AgNPs on the fiber surfaces destroyed the crystalline region of the HHK/PEO/PVA nanofibers, so that the materials further transformed from a semi-crystalline structure to an amorphous structure, which corresponded to the broader endothermic peak. At 150–240 °C, the second endothermic peak indicated the evaporation of intermolecular water from the PEO/PVA and HHK-PEO/PVA nanofibers, which was the water bound in the polymers. The addition of HHK promoted the hydrophilicity of the fibers to enhance the endothermic peak [48]. However, the endothermic peak disappeared after the silver nitrate treatment, indicating that silver replaced the water molecules to interact with HHK, PEO and PVA. This displayed that there was an essential difference between the in situ silver-anchored nanofibers and AgNP-embedded nanofibers in terms of the intermolecular interactions, which was another evidence of etched fibers treated with silver nitrate. The third endothermic peak at 250–400 °C was determined by the amide bond breaking, the amino acid residue degradation of HHK and acetyl dissociation and the complete pyrolysis of PVA [49]. The fourth endothermic peak at 400–500 °C corresponded to the thermal decomposition of the PEO and composite nanofibers [50]. Additionally, low-molecular-weight nitrogen-containing heterocyclic compounds produced by the decomposition of HHK at high temperatures crosslinked the PEO chains to greatly increase the Δ*H*_m4_ of the PEO degradation (Table 2) [51,52].

The TGA thermal analysis of the nanofibrous mats displayed three regions of weight loss (Figure 7b). The first step was due to the loss of the bound water in the mats. The second step was related to the degradation temperature of PVA and HHK at approximately 335 °C, which was consistent with the third endothermic peak that appeared in Figure 7a. AgNPs anchored on the fiber surface weakened the intermolecular force in the fibers. They could be used as the catalyst to reduce the free energy in the degradation reaction of PVA and HHK, representing that the temperature in peak 2 (Table 2) of the TGA derivative curve decreased from 335 °C to 329 °C (Table 2) [53]. The third region was near 420 °C and associated with the degradation of PEO. Similarly, the small molecular compounds produced by HHK degradation crosslinked PEO and increased the free energy of PEO degradation. The temperature in peak2 of the TGA derivative curve increased from 417 °C to 480 °C, which coincided with the results in Figure 7a. The comparison of the three nanofibrous mats in residual weight at 600 °C seemed to demonstrate that the AgNP-HHK/PEO/PVA nanofiber mat had the highest residual weight. However, it could be seen from Table 1 that the silver content in the fibers was about 10.95%, close to the 10.52% shown in Table 2. It illustrated that the organic part of the AgNP-HHK/PEO/PVA nanofiber mat was completely decomposed at 600 °C. This was beneficial for silver recycling and medical waste treatment and proved once again that AgNPs acted as a catalyst to facilitate the decomposition of PEO.

### 2.7. Antibacterial Effect of AgNPs in Nanofibrous Mats

The antimicrobial properties of nanofibrous mats against Gram-positive bacteria, Gram-negative bacteria and fungi were determined by the disk diffusion method, and the results are shown in Figure 8. The *S. aureus* and *E. coli* bacteria strains were selected as the Gram-positive and Gram-negative bacteria strains. *C. albicans* was selected as the fungi. The neat PEO/PVA and HHK/PEO/PVA nanofibrous mats displayed no antibacterial activity against the two types of bacteria and fungi, while the AgNP-HHK/PEO/PVA nanofibrous mats displayed excellent antibacterial properties. Furthermore, the AgNP-HHK/PEO/PVA nanofibrous mats were more effective in inhibiting Gram-negative bacteria compared with Gram-positive bacteria and fungi, and the inhibitory effect on the fungi was between the two. In the previous studies, AgNP was directly incorporated into the keratin/PEO/PVA nanofibers, and their antibacterial effect on the Gram-negative bacteria was also stronger than the Gram-positive bacteria [17]. The inhibition zones for the nanofibrous mats were recorded as 6.51 ± 0.10 mm, 7.33 ± 0.12 mm and 6.92 ± 0.05 mm for the Gram-positive (*S. aureus*) bacteria strain, Gram-negative (*E. coli*) bacteria and fungi (*C. albicans*). The HHK-doped PEO/PVA nanofibrous mats became a good carrier of AgNPs for wound dressings because of the more sulfur-containing groups, which might be one of the reasons why AgNP-HHK/PEO/PVA nanofibrous mats have outstanding antibacterial capabilities.

### 2.8. Cytotoxicity

Cytotoxicity is one of the most important performances for the materials used in biomedical applications. The cytotoxicity of the prepared nanofibrous mats was examined by the MTT assay in the NIH3T3 cells, according to ISO 10993-5: 2009. Generally, the samples with a cell viability larger than 75% can be considered as noncytotoxic. Figure 9 displays the cell viability of the NIH3T3 cells incubated in the extract solution from the PEO/PVA, HHK/PEO/PVA and AgNP-HHK/PEO/PVA nanofibrous mats. The 24-h, 48-h and 72-h extracts of the PEO/PVA nanofibrous mats did not show toxicity at all the dilutions and the obvious effect of promoting cell proliferation. When HHK was added into the fibers, the longer extraction time and the higher concentration of the extraction solution triggered a more significant cell proliferation. As shown in Figure 9b, the 24-h extract without dilution showed a significant effect on the cell proliferation. The 10^−2^ dilution of DMEM of the 48-h extract could also promote cell proliferation. The results indicated that HHK in the nanofibers was continuously released in DMEM, and the dissolved HHK could enhance the proliferation of the NIH3T3 cells in a dose-dependent manner. The keratin/PEO nanofibers and keratin/PVA nanofibers were proven to be nontoxic and biocompatible in the past [42,54]. Additionally, the source, species and extraction method of keratin all affected the cell proliferation [30]. It was previously reported that intracellular cysteine or its disulphide form, cystine, was related to cell proliferation, and exogenous cysteine and cystine also promoted cell proliferation [55]. After AgNP anchoring, the result was similar to Figure 9b, suggesting that the AgNPs were nontoxic to NIH3T3 cells and did not affect their proliferation. Meanwhile, the proliferation effect of the 24-h extract was significantly better than that of the HHK/PEO/PVA nanofibers. The 10^−2^ dilution of the DMEM of the 48-h and 72-h extracts also showed a significant proliferation effect, which was better than that of the HHK/PEO/PVA nanofibers. These results might be due to the fact that AgNP anchoring also dispersed the interaction of HHK and the fiber macromolecules. When the AgNPs were released, the release and dissolution of HHK were stronger than those of the HHK/PEO/PVA nanofibers without the treatment of the silver nitrate solution.

## 3. Materials and Methods

### 3.1. Materials

PEO (M_W_ = 1000 kDa), PVA (M_W_ = 77 kDa, 87–90% hydrolyzed) and silver nitrate were obtained from Sinopharm Chemical Reagent Co., Ltd. (Shanghai, China). All other chemicals and reagents used in the study were of analytic grade.

Mouse embryonic fibroblast cells (NIH3T3) were purchased from Procell Life Science & Technology Co., Ltd. (Wuhan, China).

The following are in the Appendix A: TEM images of AgNP-HHK/PEO/PVA nanofibers treated with (a) 0.005 M, (b) 0.01 M and (c) 0.02 M of silver nitrate solution and the electrospinning process and equipment.

### 3.2. Extraction of HHK

Human hair collected from Chinese barbershops was thoroughly degreased with 75% (*v*/*v*) alcohol solution and rinsed with distilled water followed by autoclaving at 15 psi. After that, 0.125-M Na_2_S was selected as the reductant to break the cystine bonds at pH 13.0 and 60 °C for 12 h, and a reduction solution was retained through the filtration. Subsequently, the filtrate was centrifuged to remove precipitation and dialyzed (MD44 dialysis membranes with a 3500-Da molecular cutoff, Solarbio, Beijing, China) in distilled water for 3 days at room temperature, changing the distilled water four times a day. The resulting extracts were lyophilized to obtain HHK for electrospinning.

### 3.3. Preparation of Spinning Solution

PEO powder, PVA powder and HHK were added to DI water at 16% (*w*/*v*), 2% (*w*/*v*) and 20% (*w*/*v*) concentrations, respectively. The obtained PEO solution, PVA solution and HHK solution were mixed with acetic acid in a volume ratio of 4:2:2:10 and then stirred evenly. In order to further improve the spinnability of the polymer solution and increase the polymer concentration, additional PEO powder was slowly added to the mixed solution during continuous stirring. Eighteen milliliters of mixed solution needed adding to 0.5-g PEO powder. Afterwards, glutaraldehyde was added to the mixture at a final concentration of 0.025% (*w*/*v*). The mixtures were stirred at room temperature for 12 h with a magnetic stirrer to ensure the complete dissolution of the polymers and obtain homogeneous solutions. The prepared solutions were left to rest for 1 h for degassing and kept in sealed containers for electrospinning.

### 3.4. Preparation of AgNP-Anchored HHK/PEO/PVA Nanofibers

The electrospinning parameters were selected as follows: 10-mL disposable sterile syringe with a 21-G flat-head stainless steel injection needle; the high DC voltage was 25 kV, provided by a high-voltage DC power supply (Dongwen High Voltage Power Supply Co., Ltd., Tianjin, China); the distance between the needle tip and collector was 15 cm; the rotating speed of the rotator drum was 700 rpm and the flow rate was 0.3 mL/h, driven by a dual-channel syringe pump (JZB-1800D, Changsha Jianyuan Medical Technology Co., Ltd., Changsha, China) (Appendix A). The electrospun nanofibers were fumigated with hydrochloric acid steam for 60 s and dried at 60 °C for 6 h to ensure their stability in the aqueous solution. A small amount (0.1 g) of the obtained nanofibers were immersed into 20 mL of 0.005-, 0.01- and 0.02-M of silver nitrate solution at room temperature overnight, then immersed in a 0.1% (*w*/*v*) NaOH solution for 24 h to neutralize the acid and washed with deionized water to remove any excessive HCl or NaOH from the nanofibers. Finally, the nanofibers were air-dried to remove any residual solvent. The nanofibers without silver were marked as HHK/PEO/PVA nanofibers, and the nanofibers without HHK and silver were marked as PEO/PVA nanofibers. 

### 3.5. In Vitro Degradation of Nanofibers

The degradation of the AgNP-anchored HHK/PEO/PVA nanofibers was observed with a transmission electron microscope (TEM; Talos F200X G2, Thermo Fisher Scientific, Waltham, MA, USA) at a voltage of 200 kV by the analysis of the fine structure and element distribution of the nanofibers. All specimens treated with silver nitrate were immersed in ethanol and ultrasonicated at 300 W for 10 min, followed by washing with water and dispersing. The suspension of AgNP-anchored HHK/PEO/PVA nanofibers was deposited on a sample grid (carbon membrane supported by a copper grid) and allowed to dry for 2 h before TEM observation. In addition, all specimens treated with silver nitrate were immersed in ethanol and ultrasonicated at 300 W and then washed and dried at a certain interval to weigh and calculate the degradation rate.

### 3.6. Characterizations

The surface morphology of the manufactured nanofibers was determined by a scanning electron microscope (SEM; Gemini SEM 300, Carl Zeiss, Jena, Freistaat Thüringen, Germany). All specimens were coated with platinum under an EM ACE600 High-Vacuum Sputter Coater (Leica, Weztlar, Hesse-Darmstadt, Germany) for 60 s before SEM observation. Image analysis software (ImageJ, version 1.5.2) was used to calculate the average diameters of the nanofibers by taking 50 readings randomly for each sample.

The chemical structures of the nanofiber samples were assayed on a Fourier-transform infrared (FTIR) spectrometer (TENSOR 27, Bruker Co., Karlsruhe, Baden-Württemberg, Germany). All spectra were collected by transmission mode in the wavelength range of 4000–400 cm^−1^ in a dry atmosphere at room temperature to observe the possibility of any chemical interactions among the AgNPs, kerateine and PEO/PVA.

X-ray diffraction (XRD) was used to analyze the crystallization characteristics of the prepared nanofibers using a nickel-filtered Cu-Ka radiation-assisted Multipupose X-ray diffractometer (Ultima IV, Rigaku Co., Osaka, Japan). The scanning speed was kept at 5°/min, and the 2θ range was located from 5° to 80°.

The combination of thermogravimetry (TGA) and differential scanning calorimetry (DSC) of the nanofibers was performed by a thermogravimetric analysis using a STA 449 F5 synchronous thermal analyzer (NETZSCH Co., Selb, Bavaria, Germany). It was operated in static mode under atmospheric air at a heating rate of 10 °C/min and a temperature range of 30–600 °C.

### 3.7. Antibacterial Activity Test

*Staphylococcus aureus* ATCC 6538, *Escherichia coli* ATCC 8739 and *Candida albicans* ATCC 10231 were used as test the bacteria to assess the antibacterial properties of the nanofibrous mats against Gram-positive bacteria, Gram-negative bacteria and fungi.

Typically, the bacterial and fungal strains were inoculated into a nutrient broth medium and yeast extract–peptone–dextrose medium, respectively, and then shaken at 37 °C for 16–24 h until the OD_600 nm_ of the fermentation broth reached 0.6. Twenty microliters of fermentation broth were taken to cover the agar plate. The nanofibrous mats cut into a 5-mm diameter wafer using a punch were sterilized under UV irradiation overnight and then placed on the agar plate to culture overnight at 37 °C. The diameter of the inhibition zone was determined.

### 3.8. In Vitro Cytotoxicity Test

The NIH3T3 mouse embryo fibroblast cell line was used for studying the in vitro effect of the nanofibrous mats on the cell viability by the MTT assay. The cytotoxicity test of the samples was carried out according to ISO 10993-5. The cells were maintained in Dulbecco’s modified Eagle’s medium (DMEM) with 10% fetal bovine serum (Gibco, Shanghai, China), 100-units/mL penicillin, 100-μg/mL streplomycin (HyClone Laboratories, Inc., USA) under atmospheric pressure and 5% carbon dioxide humidification at 37 °C. Cells were seeded onto 96-well plates with a seeding density of 4.0 × 10^4^ cells/mL in a well for 24-h incubation. The PEO/PVA, HHK/PEO/PVA and AgNP-HHK/PEO/PVA nanofibrous mats were punched into 10-mm diameter discs. The sample discs were sterilized by ultraviolet irradiation for 30 min and soaked in 10-mL DMEM for 24 h, 48 h and 72 h to obtain the extract solution. The resulting solution was diluted with DMEM at different concentrations and replaced the old medium in the well. After 24 h of total culture, the medium was discarded, and the environmental safety and biosafety of the products released from the AgNP-HHK/PEO/PVA nanofibrous mats were assessed by the MTT assay. The OD value at 570 nm was measured with a microplate reader (Synergy2, BioTek Instruments, Inc., Winooski, VT, USA).

### 3.9. Statistical Analysis

The mean and standard deviation (SD) were used to describe the data. The data were analyzed using ANOVA, and then, an unpaired *t*-test was used to analyze multiple differences between the groups by using GraphPad Prism software, version 8. A *p*-value at less than 0.05 was considered statistically significant.

## 4. Conclusions

AgNPs was successfully anchored on the electrospun HHK/PEO/PVA nanofiber mats. It was confirmed by TEM-EDS that most of AgNPs were bounded to the sulfur-containing amino acid residues of HHK. Due to the acid environment derived from the silver ion reduction and the subsequent alkali treatment, the nanofiber surface was etched, and the fiber diameter decreased. The optimal concentration of silver nitrate was 0.005 M, according to the results of the antidegradation. The FTIR, XRD and thermal analyses showed that the hydrogen bonds between PEO and PVA were broken by HHK and further by the AgNPs. The AgNPs could accelerate the degradation of the macromolecules in the fibers under the heating conditions. However, the performances of the AgNP-HHK/PEO/PVA nanofibrous mats were satisfactory in the two target applications of antibacterial and biocompatibility that we focused on. It can be concluded that the AgNP-HHK/PEO/PVA nanofibrous mats are considered as candidate biomedical materials on account of their excellent antibacterial activity against both Gram-positive and Gram-negative bacteria and cell proliferation-promoting properties.

## Figures and Tables

**Figure 1 molecules-26-02783-f001:**
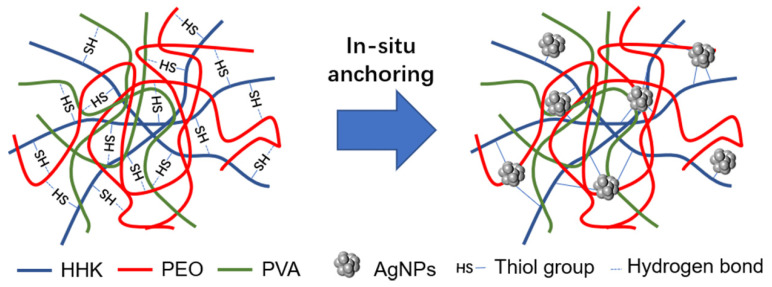
Schematic illustration of in situ anchoring.

**Figure 2 molecules-26-02783-f002:**
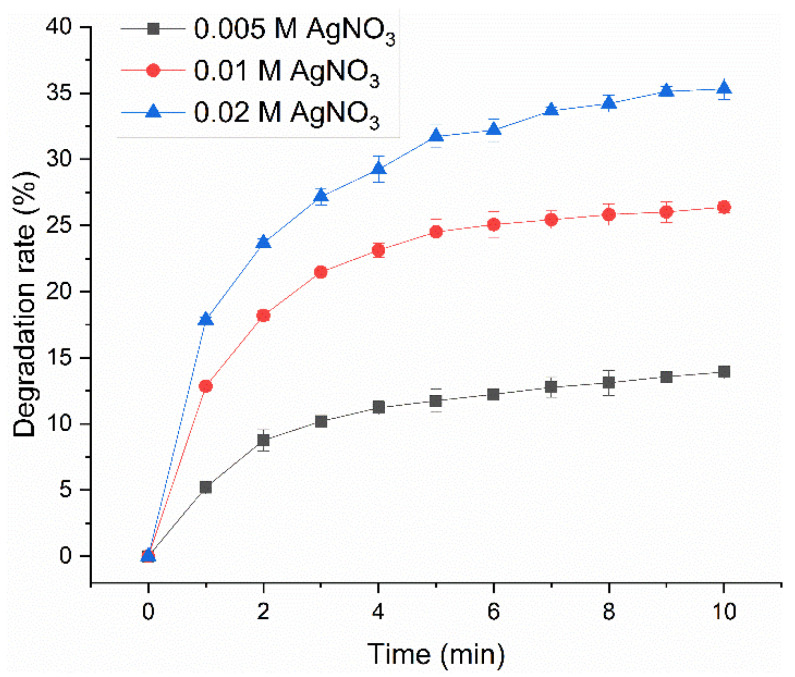
Degradation behavior of the HHK/PEO/PVA nanofibers treated with different concentrations of the silver nitrate solution.

**Figure 3 molecules-26-02783-f003:**
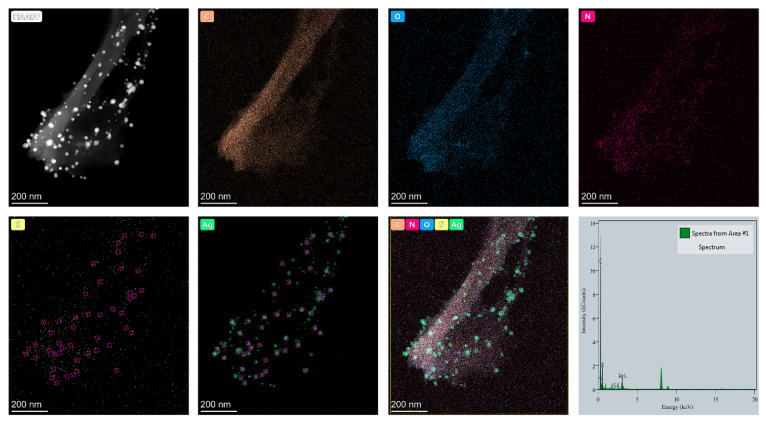
HAADF image, EDS mapping and an analysis of the AgNP-HHK/PEO/PVA nanofibers.

**Figure 4 molecules-26-02783-f004:**
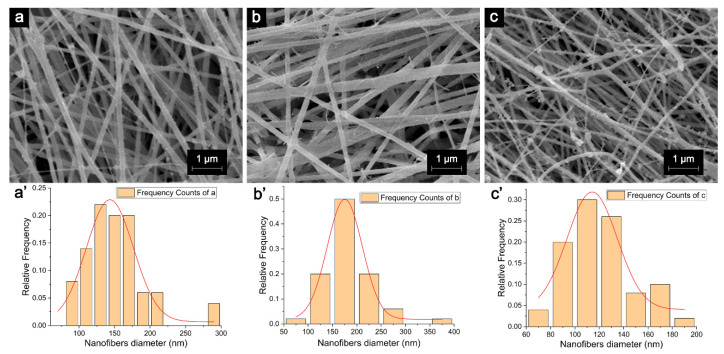
SEM micrographs and diameter distribution histograms of (**a**,**a′**) PEO/PVA nanofibrous mats, (**b**,**b′**) HHK/PEO/PVA nanofibrous mats and (**c**,**c′**) AgNP-HHK/PEO/PVA nanofibrous mats.

**Figure 5 molecules-26-02783-f005:**
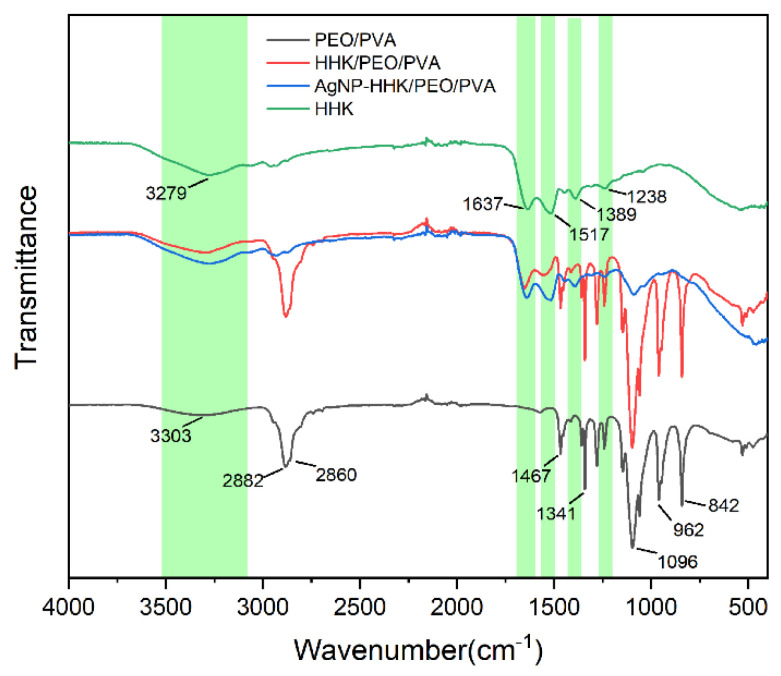
FTIR spectra of the PEO/PVA, HHK/PEO/PVA and AgNP-HHK/PEO/PVA nanofibrous mats.

**Figure 6 molecules-26-02783-f006:**
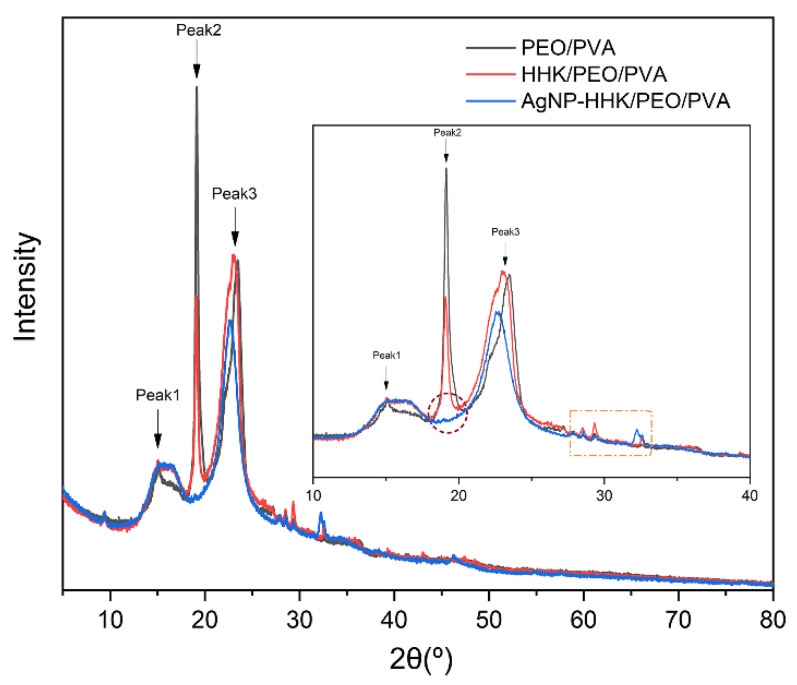
X-ray diffraction patterns of the PEO/PVA, HHK/PEO/PVA and AgNP-HHK/PEO/PVA nanofibrous mats.

**Figure 7 molecules-26-02783-f007:**
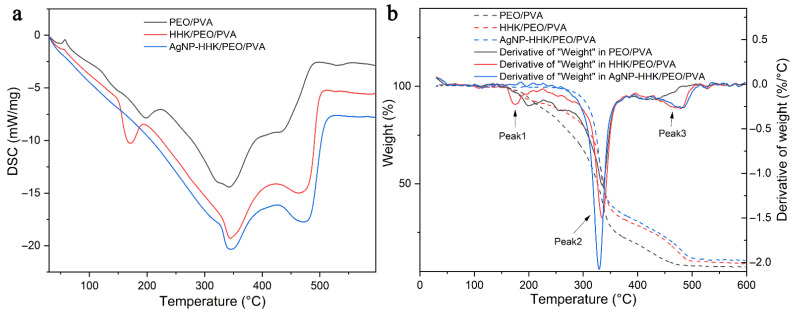
(**a**) DSC curves and (**b**) TGA and TGA derivative curves of the PEO/PVA, HHK/PEO/PVA and AgNP-HHK/PEO/PVA nanofibrous mats.

**Figure 8 molecules-26-02783-f008:**
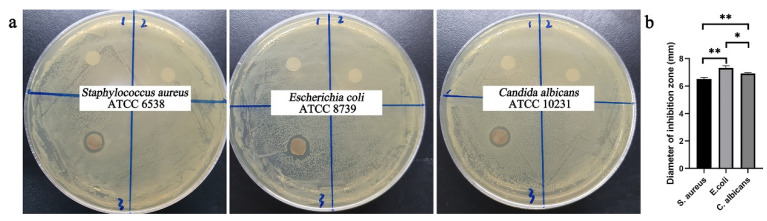
(**a**) Inhibition zones for neat PEO/PVA (1), HHK/PEO/PVA (2) and AgNP-HHK/PEO/PVA (3) nanofibrous mats against *S. aureus*, *E. coli* and *C. albicans*, and (**b**) the results analysis of the antibacterial test. *, *p* < 0.05; **, *p* < 0.01; *n* = 3.

**Figure 9 molecules-26-02783-f009:**
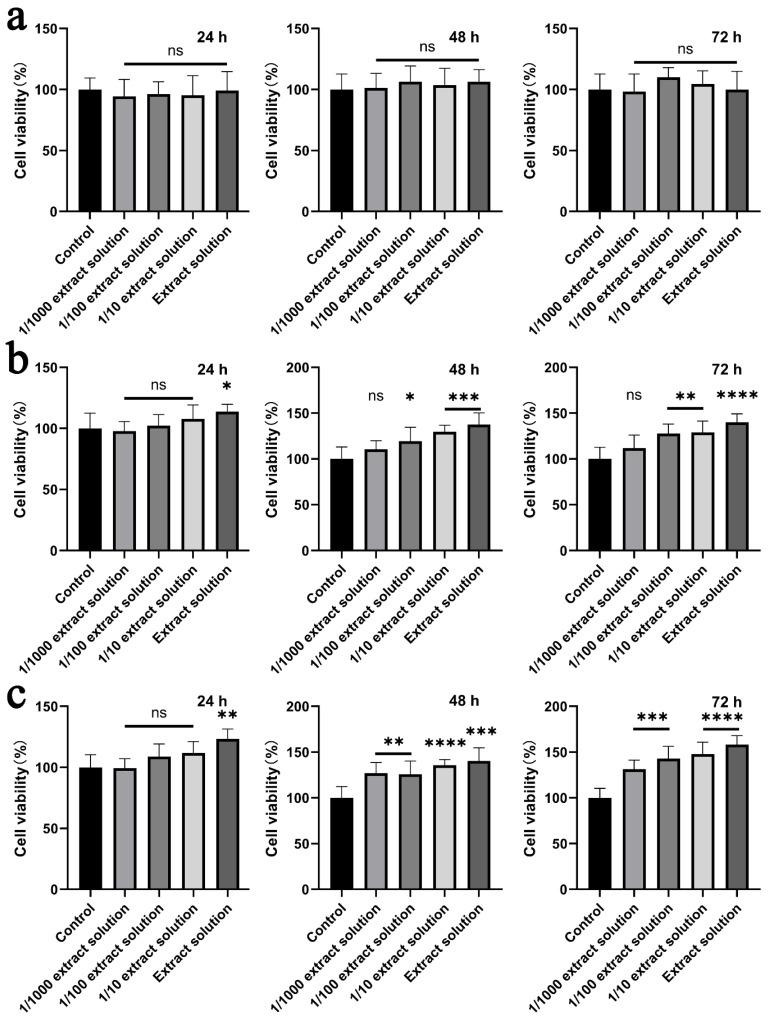
Cytotoxicity profile of the solution extracted from the (**a**) PEO/PVA, (**b**) HHK/PEO/PVA and (**c**) AgNP-HHK/PEO/PVA nanofibrous mats by the NIH3T3 cells. *, *p* < 0.05; **, *p* < 0.01; ***, *p* < 0.001; ****, *p* < 0.0001. ns, *p* > 0.05, which meant that there were no significant differences compared to the control; *n* = 6.

**Table 1 molecules-26-02783-t001:** Element analysis of the spectra from Area #1 (Figure 3).

Element	Atomic Fraction (%)	Atomic Error (%)	Mass Fraction (%)	Mass Error (%)	Fit Error (%)
C	90.02	3.38	79.01	1.88	0.49
O	8.17	1.67	9.55	1.93	1.73
N	0.38	0.21	0.39	0.22	51.31
S	0.04	0.01	0.09	0.02	4.08
Ag	1.39	0.17	10.95	1.28	0.13

**Table 2 molecules-26-02783-t002:** DSC and TGA data of the (a) PEO/PVA, (b) HHK/PEO/PVA and (c) AgNP-HHK/PEO/PVA nanofibrous mats.

Sample	DSC								TGA			
	*T*_m1_ /°C	Δ*H*_m1_/J·g^−1^	*T*_m2_ /°C	Δ*H*_m2_/J·g^−1^	*T*_m3_ /°C	Δ*H*_m3_/J·g^−1^	*T*_m4_ /°C	Δ*H*_m4_/J·g^−1^	Peak1 /°C	Peak2 /°C	Peak3 /°C	Residue at 600 °C/%
a	47.57	−52.88	195.96	−621.34	339.29	−3256.72	439.06	−303.48	199.70	335.00	417.72	7.31
b	44.93	−29.68	167.25	−491.06	346.11	−2909.37	481.08	−2153.35	174.84	336.32	479.54	9.01
c	39.87	−44.34	None	None	343.45	−2687.29	481.39	−2421.50	None	329.39	482.18	10.52

## Data Availability

Not applicable.

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
