# Peer review of "Silver Nanoparticle-Anchored Human Hair Kerateine/PEO/PVA Nanofibers for Antibacterial Application and Cell Proliferation"

_molecules, 2021, doi:10.3390/molecules26092783_

Round 1

Reviewer 1 Report

The paper entitled “Silver nanoparticle-anchored human hair kerateine/PEO/PVA nanofibers for antibacterial application and cell proliferation” by Tang et al. deals with the preparation and characterization of different electrospun fibers with potential antibacterial properties for wound healing.

The manuscript is clear, well written and the conclusions are supported by the results. However, some very minor corrections are needed:

  1. Please indicate the hydrolysis degree of the PVA sample
  2. In fig. 8, the cell viability is usually expressed in percent (%).
  3. Why the authors have used a PEO with a Mw = 1000 g/mol?! It would have been of interest to study the influence of the molar mass of PEO on the electrospun fiber’s characteristics.

Author Response

Reviewer 1:

The paper entitled “Silver nanoparticle-anchored human hair kerateine/PEO/PVA nanofibers for antibacterial application and cell proliferation” by Tang et al. deals with the preparation and characterization of different electrospun fibers with potential antibacterial properties for wound healing.

The manuscript is clear, well written and the conclusions are supported by the results. However, some very minor corrections are needed:

Please indicate the hydrolysis degree of the PVA sample

In fig. 8, the cell viability is usually expressed in percent (%).

Why the authors have used a PEO with a Mw = 1000 g/mol?! It would have been of interest to study the influence of the molar mass of PEO on the electrospun fiber’s characteristics.

Dear reviewer:

Thanks for your comments and suggestions. After a careful review and analysis of this study, our responses to the reviewer’s suggestions are as follows:

Comment 1: Please indicate the hydrolysis degree of the PVA sample

Response (page 14, line 353): Thank you for your suggestion. We have added the hydrolysis degree of the PVA sample, “87-90% hydrolyzed”.

Comment 2: In fig. 8, the cell viability is usually expressed in percent (%).

Response (page 13, Figure 9): Thank you for your suggestion. We have corrected the expression of the cell viability in Figure 9. All cell viabilities are expressed in percent (%).

Comment 3: Why the authors have used a PEO with a Mw=1000 g/mol?! It would have been of interest to study the influence of the molar mass of PEO on the electrospun fiber’s characteristics.

Response: Thank you for your suggestion. We used PEO (Mw=1000 kDa) as the substrate of electrospinning nanofibers mainly based on the dual consideration of fiber formation and solubility. PEO with large molecular weight has longer molecular chain and better fiber-forming property, but its solubility is poor and the viscosity of PEO solution is also high. In this study, two-step addition of PEO was used to dissolve and mix PEO. PEO with small molecular weight has high solubility, but its fiber-forming property is poor. Considering comprehensively, we finally chose PEO (Mw=1000 kDa). Of course, the influence of molecular weight of PEO on electrospun fibers is fundamental. According to the literature (Filip, P and peer, P. characterization of poly (ethylene oxide) nanofibers mutual relations between mean diameter of electrospun nanofibers and solution characteristics. Processes, 2019,7 (12): 948), the molecular weight of PEO affected the fiber diameter and the viscosity of PEO solution. Under the same PEO concentration, the diameter of electrospun PEO fibers increased with the increased molecular weight. The fibers with larger diameter and polymers with higher molecular weight also had better resistance to degradation.

Kind regards,

Yours sincerely,

Yan Ge

School of Textile and Clothing, Nantong University, Nantong 226019, PR China

National & Local Joint Engineering Research Center of Technical Fiber Composites for Safety and Protection, Nantong University, Nantong 226019, PR China

Reviewer 2 Report

This work is devoted to studying the possibilities of medical application of a composite material based on keratin and silver nanoparticles with polymer fillers. The authors carried out an extensive study of the physicochemical and biological properties of the obtained material using a variety of modern methods.
However, this work leaves a number of questions. For example, are there any advantages of using anchored silver nanoparticles compared to materials obtained by mixing polymers with pre-prepared silver nanoparticles (as was done in the works of other authors)? The authors point out that the use of a solution of 0.005M silver nitrate does not lead to degradation of polymer fibers, in contrast to solutions with a higher concentration. Has the kinetics of this process been studied? Is it possible that longer exposure to 0.005M silver nitrate still degrades the material?

Author Response

Reviewer 2:

This work is devoted to studying the possibilities of medical application of a composite material based on keratin and silver nanoparticles with polymer fillers. The authors carried out an extensive study of the physicochemical and biological properties of the obtained material using a variety of modern methods.
However, this work leaves a number of questions. For example, are there any advantages of using anchored silver nanoparticles compared to materials obtained by mixing polymers with pre-prepared silver nanoparticles (as was done in the works of other authors)? The authors point out that the use of a solution of 0.005M silver nitrate does not lead to degradation of polymer fibers, in contrast to solutions with a higher concentration. Has the kinetics of this process been studied? Is it possible that longer exposure to 0.005M silver nitrate still degrades the material?

Dear reviewer:

Thanks for your comments and suggestions. After a careful review and analysis of this study, our responses to the reviewer’s suggestions are as follows:

Comment 1: are there any advantages of using anchored silver nanoparticles compared to materials obtained by mixing polymers with pre-prepared silver nanoparticles (as was done in the works of other authors)?

Response: Compared with the materials obtained by mixing the polymer with the pre-prepared silver nanoparticles, we used the nano-silver treatment of fiber surface similar to that of other researchers. First of all, we extracted the reduced-type human hair keratin (kerateine). From the co-localization of sulfur and silver in TEM, we confirmed that protein molecules with a large number of sulfur groups provided active sites for the AgNPs anchoring on the fiber surface. Secondly, most of the silver was wrapped in the fibers prepared by mixing polymers with silver nanoparticles. It was difficult for silver to migrate from fiber interior to fiber surface, which might lead to the insufficient sustainable antibacterial properties of blended fibers or the waste of silver nanoparticles. In this study, 10% weight of nanofibers we prepared was AgNPs (according to TGA data), and all AgNPs were covered on the surface of fibers, which could maximize the antibacterial performance.

Comment 2: The authors point out that the use of a solution of 0.005M silver nitrate does not lead to degradation of polymer fibers, in contrast to solutions with a higher concentration. Has the kinetics of this process been studied?

Response (page 3, lines 103-111, lines114-119; page 4, Figure 2 and page 15, lines 400-402): In this study, the anti-degradation behavior of nanofibers treated with different concentrations of silver nitrate solution was compared and analyzed. It was concluded that the fibers treated with 0.005 M silver nitrate solution were more tolerant to the dual degradation caused by ultrasound and ethanol than those treated with higher concentration of silver nitrate solution. It was different from "the use of a solution of 0.005 M silver nitrate does not lead to degradation of polymer fibers, in contrast to solutions with a high concentration" understood by the reviewer. We also supplemented the degradation kinetics results of nanofibers treated with different concentrations of silver nitrate solution, as shown in Figure 2. It was found that the results were consistent with the previous judgment. The higher concentration of silver nitrate led the faster degradation rate of fibers.

Comment 3: Is it possible that longer exposure to 0.005M silver nitrate still degrades the material?

Response: We compared the degradation of AgNPs-HHK/PEO/PVA nanofibers treated with different concentrations of silver nitrate solution under extreme conditions. We found that the higher concentration of silver nitrate solution led the more serious degradation of fibers. Therefore, we think that the degradation of materials must be closely related to silver nitrate. Longer exposure to 0.005 M silver nitrate will definitely degrade the materials, and the degradation rate is lower than that treated with high concentration of silver nitrate. Based on the above analysis, the soaking time of materials in silver nitrate and sodium hydroxide solution must be carefully controlled to avoid unnecessary degradation. However, the exposure to silver nitrate solution was only a treatment step, not an application step of materials, so we did not discuss it in detail in the article.

Kind regards,

Yours sincerely,

Yan Ge

School of Textile and Clothing, Nantong University, Nantong 226019, PR China

National & Local Joint Engineering Research Center of Technical Fiber Composites for Safety and Protection, Nantong University, Nantong 226019, PR China

Reviewer 3 Report

This paper suggests the development of Silver nanoparticle-anchored human hair kerateine/PEO/PVA 2 nanofibers for antibacterial application and cell proliferation. It is an interesting and well-written paper. A good range of characterisations has been done which make the paper suitable for publications in Molecules. Below are some minor suggestions

  • More details on the electrospinning parameters and equipment used for each experiment should be provided.
  • How would the nanofiber diameter affect the Antibacterial performance of the nanofiber mat?
  • Fonts in Figure 3 are hardly readable (particularly in SEM images). Please enhance the quality of the images.
  • Section 2.8 should be expanded with more details.

Author Response

Reviewer 3:

This paper suggests the development of Silver nanoparticle-anchored human hair kerateine/PEO/PVA 2 nanofibers for antibacterial application and cell proliferation. It is an interesting and well-written paper. A good range of characterisations has been done which make the paper suitable for publications in Molecules. Below are some minor suggestions

  • More details on the electrospinning parameters and equipment used for each experiment should be provided.
  • How would the nanofiber diameter affect the Antibacterial performance of the nanofiber mat?
  • Fonts in Figure 3 are hardly readable (particularly in SEM images). Please enhance the quality of the images.
  • Section 2.8 should be expanded with more details.

Dear reviewer:

Thanks for your comments and suggestions. After a careful review and analysis of this study, our responses to the reviewer’s suggestions are as follows:

Comment 1: More details on the electrospinning parameters and equipment used for each experiment should be provided.

Response (Figure S2 in supplementary materials and page14, lines 380-386): We have added the relevant electrospinning parameters and equipment details in article, and inserted the photo of electrospinning process(Figure S2)in the supplementary file.

Comment 2: How would the nanofiber diameter affect the Antibacterial performance of the nanofiber mat?

Response: The results of antibacterial test showed that HHK/PEO/PVA nanofibers treated with silver nitrate had antibacterial properties, while the untreated PEO/PVA and HHK/PEO/PVA nanofibers did not exhibit obvious antibacterial effect. The results indicated that the antibacterial properties of fibers were attributed to the AgNPs anchored on their surface. The amount of AgNPs in the nanofibers could affect the antibacterial property and durability of nanofibrous mats. The antibacterial experiment in this study could not get the relevant results about the effect of nanofiber diameter on the antibacterial properties of nanofibrous mats. Generally speaking, the smaller fiber diameter led the larger specific surface area for nanofibrous mats with the same mass to increase the anchored AgNPs, which was obviously beneficial to the sustained antibacterial activity. On the contrary, the larger diameter decreased the sustained antibacterial effect of nanofibers. However, the diameter of AgNPs-HHK/PEO/PVA nanofibers was not effectively controlled in this study, so this hypothesis could not be discussed and verified in detail.

Comment 3: Fonts in Figure 3 are hardly readable (particularly in SEM images). Please enhance the quality of the images.

Response (page 6, Figure 4): We have improved the image quality and make it readable by optimizing the scale and font size of Figure 4.

Comment 4: Section 2.8 should be expanded with more details.

Response (pages 10-11, lines 323-324, 326-330, 332-334 and 338-345): We extended the analysis of results in section 2.8. The effect of AgNPs on cell proliferation of HHK dissolved from nanofibers was concretely analyzed. The statistical results of cell proliferation were analyzed simultaneously in detail.

Kind regards,

Yours sincerely,

Yan Ge

School of Textile and Clothing, Nantong University, Nantong 226019, PR China

National & Local Joint Engineering Research Center of Technical Fiber Composites for Safety and Protection, Nantong University, Nantong 226019, PR China

Round 2

Reviewer 2 Report

The authors answered all my questions, so I believe that this article can be published in the presented form